# TEG-INSTRUCT: TOWARDS PREMIUM INSTRUCTION-TUNING DATA VIA TEXT-GROUNDED TASK DESIGN

## ABSTRACT

The enhancement of language model capabilities heavily relies on the availability of high-quality instruction-tuning data. However, current data collection approaches face problems due to the high costs associated with manual labeling or the hallucination of relying solely on LLMs. To overcome these problems, this paper proposes a scalable solution for automatically gathering top-notch instruction-tuning data. Our proposed method involves training LLMs to autonomously design tasks utilizing human-written texts, thereby aiding LLMs in mitigating erroneous outputs. In contrast to other methods that utilize human-written texts, our method employs a task generator capable of simultaneously producing the instruction, input, and output. It aims to minimize the introduction of noise from the original text and ensure coherent and aligned task components. Additionally, we train a discriminator to identify and filter out invalid tasks that might contain hallucination, thus further improving the quality of the collected data. The results of the automated and manual evaluation experiments validate the reliability and validity of our proposed dataset, demonstrating the applicability it brings to LLMs for both in-domain and out-of-domain generation.

## 1 INTRODUCTION

Recent efforts in the NLP community have focused on *instruction-tuning* (Sanh et al., 2022; Mishra et al., 2022; Wei et al., 2022), i.e., improving large language models' (LLMs) (Brown et al., 2020; Chowdhery et al., 2022; Touvron et al., 2023a) capacity to understand and follow instructions. Advanced LLMs have been trained to be capable of generating customized outputs when provided with specific instructions (with inputs), enabling them to adapt to new tasks without prior exposure.

As a crucial problem in improving the instruction-following capability, how to collect high-quality instruction-tuning data is gaining popularity. Previous methods can be broadly divided into three primary categories: a) SUPER-NI (Wang et al., 2022) and DOLLY (Conover et al., 2023) hire professionals to write instructions for diverse NLP tasks, whereas are limited in scale due to the labor-intensive nature of the annotation process. b) SELF-INSTRUCT (Wang et al., 2023) and ALPACA (Taori et al., 2023) advocate the use of LLMs to automatically generate instructions to eliminate labor costs. However, their outputs are plagued by the hallucination problems (Zhang et al., 2023; Zheng et al., 2023) of LLMs, which leads to poor data quality. In fact, only 54% data in SELF-INSTRUCT is valid (Wang et al., 2023). Furthermore, the diversity of instructions may be constrained by the knowledge limitations of LLMs. c) Dynosaur (Yin et al., 2023) employs LLMs to convert the existing NLP datasets from Huggingface[1] to instruction-tuning data at a lower API cost. Unfortunately, it is not applicable to the scenarios where no labeled data is available.

Recent research (Köksal et al., 2023; Li et al., 2023) has provided a more potential idea: using human-written text as gold responses for user instructions and utilizing LLMs to predict the instructions. It is called *instruction back-translation* (Li et al., 2023). In this way, massive instruction-tuning data can be easily constructed and somewhat alleviate the hallucinations of the outputs. However, we believe that these methods still face two realistic challenges that may limit the quality of generated data: a) Text may contain noise that is not suitable as a gold response. Figure 1 gives a bad case from the LONGFORM (Köksal et al., 2023) dataset, where the human-written text contains

---

[1]https://huggingface.co/

Figure 1: Differences between our proposed dataset TEG-INSTRUCT with previous LONGFORM (similar with HUMPBACK dataset). Red text indicates noise that is not related to the generated task. Blue text indicates output that has been touched up by LLM to be more consistent with the task form. Underlined is the input field separated from the instruction.

noise (red). The noisy part is not related to the generated task but incorrectly added to the response (yellow box) without any refinement. b) Lack of input field is detrimental to LLM's generalization capability in diversity tasks. As shown in Figure 1, the existing datasets have only an instruction field and an output field but no input field. We hypothesize that adding inputs can decouple the overly complex instructions, thus allowing LLMs to more accurately perceive similar tasks.

In this paper, we propose a novel paradigm for collecting instruction-tuning data by training LMs as task designers to automatically generate the instruction, input, and output based on the given human-written text. We call the resulting dataset TExt-Grounded Instructions (TEG-INSTRUCT), containing 270k instruction data. Briefly, the collection of TEG-INSTRUCT consists of two phases. In the first stage, we utilize a well-aligned LLM as an advanced teacher to build a meta-training set. To make it applicable to human texts and ensure diversity in the generated tasks, we leverage in-context learning but sample demonstrations from two distinct sources: the real human-written corpus and the existing instruction-tuning dataset. In the second stage, we train two LMs as a task generator and a task discriminator, respectively. The former aims to design tasks based on the given text, while the latter evaluates the designed tasks in order to retain high-quality examples. Unlike conventional methods, our method is grounded in text, while leveraging LMs for integration and refinement, resulting in data of superior quality. In addition, given that our used language models are fully accessible, they can be effectively utilized to generate instruction-tuning data for scenarios where data privacy is essential. Through extensive manual evaluation, we have demonstrated that our TEG-INSTRUCT method effectively addresses input and output hallucination issues while reducing noise derived from the original text. Furthermore, automatic evaluation results indicate that LLMs trained using TEG-INSTRUCT maintain competitive instruction-following performance on both in-domain and out-of-domain datasets. In summary, the contributions of this paper include:

- We release a high-quality text-grounded instruction-tuning dataset, which comprises 270k examples consisting of (instruction, input, output) triples.

- We propose a novel paradigm for collecting instruction-tuning data by training LLMs to automatically design tasks based on the given human-written text to effectively aid in reducing hallucination within LLMs. Moreover, since our used LLM is open-source, it has the potential to generate vast (10 million level or more) amounts of diverse instruction-tuning data.

- We conduct a comprehensive manual and automated evaluation and demonstrated that our dataset is of high quality and improves in-domain and out-of-domain capabilities for LLMs.

## 2 PROBLEM FORMULATION

Given a corpus of human-written documents $\mathcal{C} = \{\mathcal{D}_1, \mathcal{D}_2, ..., \mathcal{D}_n\}$, the text-grounded instruction-tuning data collection aims to build a task designer $\mathcal{M}$. For each document $\mathcal{D}_i \in \mathcal{C}$, $\mathcal{M}$ returns a valid task $\mathcal{T}_i := \mathcal{M}(\mathcal{D}_i)$ when $\mathcal{D}_i$ is *self-contained*, otherwise null. Here, *self-contained* means that $\mathcal{D}_i$ contains all the information needed to construct at least one complete task. The valid task

$\mathcal{T}_i = (\mathcal{P}_i, \mathcal{I}_i, \mathcal{O}_i)$ is an instruction-tuning instance, where $\mathcal{P}_i$, $\mathcal{I}_i$ and $\mathcal{O}_i$ denote the instruction field, input field and output field, respectively. Ideally, the document $\mathcal{D}_i$ can belong to any domain or style. This paper attempts six types of semi-structured and unstructured text, including Wikipedia, academic papers, code, etc (see Section 3.1 for details). Eventually, after all documents in $\mathcal{C}$ have been processed, an instruction-tuned dataset $\mathcal{T} = \{\mathcal{T}_1, \mathcal{T}_2, ..., \mathcal{T}_m\}$ is available, where $m \leq n$.

## 3    Collection of TeG-Instruct Data

Figure 2 shows the construction process of TeG-Instruct, which can be divided into two phases. In the first phase, sampled texts are annotated using ChatGPT to create a text-relevant task. After undergoing post-processing, a meta-training set comprising pairs of (document, task) is obtained. In the second phase, we utilize the meta-training set to train a task generator and a task discriminator.

### 3.1    Corpus & Document Sampling

We start with the Pile (Gao et al., 2021), a multi-domain, multi-style human-written text corpus. Specifically, we choose six diverse corpora in Pile, namely ArXiv, FreeLaw, StackExchange, Wikipedia, Github, and DeepMind Mathematics (DM Math). To ensure that each sampled document is as self-contained as possible, we design specific sampling methods for different corpora. For longer documents such as ArXiv and FreeLaw, successive segments are randomly intercepted within the character range of 2000 to 3500; For shorter documents such as DM Math, a question-answer pair is randomly selected; For Wikipedia, Github, and StackExchange, the entire document is used directly. Note that the sampled document may have the potential to generate more than one task, but this is not the focus of this paper and left for future work.

### 3.2    Meta-Training Set Construction with Dual-View ICL

To train an ideal task designer, we first need to collect high-quality training examples of document-task pairs. Inspired by (Li et al., 2023), we leave this job to ChatGPT (OpenAI, 2023) to reduce labor costs. Concretely, for each sampled document $\mathcal{D}_i$, we ask ChatGPT to create a task $\mathcal{T}_i$ relevant to $\mathcal{D}_i$ by simultaneously predicting the instruction, input, and output. To accomplish this goal, we harness the power of *in-context learning* (ICL). In particular, the prompt fed to ChatGPT is denoted by $(\mathcal{P}^*, x_1, ..., x_k)$. The full prompt is shown in Appendix A. Here $\mathcal{P}^*$ is the instruction that describes our goal and $x_i = (\mathcal{D}_i, \mathcal{T}_i)$ represents $i$-th demonstration sampled from a seed set $\mathcal{X}$. In contrast to previous methods like Self-Instruct that started with only manually annotated tasks as seed sets, we let $\mathcal{X} = \mathcal{X}_d \cup \mathcal{X}_t$ contain examples from two different views.

**Document-View Seed Examples** $\mathcal{X}_d$**.** The examples in this view are designed to allow ChatGPT to quickly adapt to the style of the target text when creating tasks. For each corpus, 20 pairs (document, task) are first manually constructed as the initial seeds $\mathcal{X}_d^0$. Subsequently, 200 documents are sampled and each is wrapped with $\mathcal{P}^*$ and 5 randomly selected demonstrations from $\mathcal{X}_d^0$, resulting in 200 prompts. Finally, ChatGPT takes these prompts as inputs and returns 200 tasks. After manual checking, the retained valid document-task pairs, denoted by $\mathcal{X}_d$, characterize the target text style and the tasks that are appropriate to design based on that text.

**Task-View Seed Examples** $\mathcal{X}_t$**.** Recent studies (Sanh et al., 2022; Li et al., 2023) have reached a consensus that diverse instructions are necessary for LLMs. However, we hypothesize that it is difficult to ensure diversity by relying only on a few dozen manually constructed initial seeds in $\mathcal{X}_d$. To further increase the diversity of tasks, we propose to leverage the examples from the existing instruction-tuning dataset to inspire the process. Initially, we sample 50 tasks from the Alpaca-GPT4 dataset. Notice that there are only tasks and no corresponding documents since these tasks are automatically generated by the ChatGPT. Therefore, we perform an inverse process to transform these 50 tasks into possible human-written documents. In our experiments, this work is also done by ChatGPT using ICL. The procedure is similar to generating document-view seeds, except that we manually construct another 20 initial seeds $\mathcal{X}_t^0$ based on Alpaca. The resulting 500 pairs (document, task) are denoted by $\mathcal{X}_t$ and provide ChatGPT with the potential for more diverse tasks.

**Post-Processing.** In fact, despite the use of dual-view ICL, the tasks generated by ChatGPT are not always valid due to the *hallucination* problem. Thus, we propose a simple but efficient post-

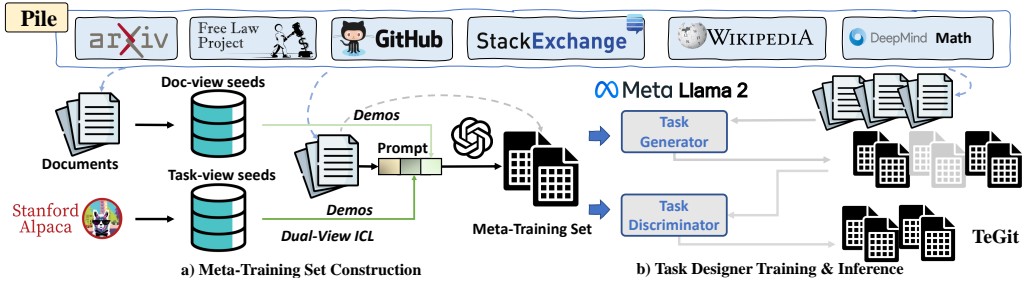

Figure 2: Overview of TEG-INSTRUCT construction process.

processing method to filter invalid examples. A natural idea is to use the similarity between instructions, inputs, and outputs and a given document to determine whether a task is valid. Motivated by this, we devise a score $\sigma(\mathcal{T}_i) = \min(\tilde{\sigma}(\mathcal{D}_i, \mathcal{I}_i), \tilde{\sigma}(\mathcal{D}_i, \mathcal{O}_i))$, where $\tilde{\sigma}(\mathcal{D}_i, s) = |t(\mathcal{D}_i) \& t(s)|/|t(s)|$ and $t(s)$ denotes the set of tokens of $s$. All examples $(\mathcal{D}_i, \mathcal{T}_i)$ will be removed where $\sigma(\mathcal{T}_i) < \theta$. In addition, in a valid task, the inputs and outputs must follow the instruction $\mathcal{P}_i$. To verify this, we use ChatGPT to determine if a task makes logical sense by actually completing it. First, we concatenate $\mathcal{P}_i$ and $\mathcal{I}_i$ as a prompt $\mathcal{Z}$ and delete this example if ChatGPT is unable to answer. Subsequently, we add $\mathcal{D}_i$ to $\mathcal{Z}$ as a new prompt to ask ChatGPT and compute the similarity of ChatGPT's reply to the text previously generated output. The examples where $\sigma$ is smaller than the threshold $\theta$ are removed. In the end, we obtain the *meta-training set* $\Omega = \Omega^+ \cup \Omega^-$, where $\Omega^-$ denotes the examples removed by post-processing and $\Omega^+$ denotes the retained ones.

## 3.3 TEXT-GROUNDED TASK DESIGNER

In this stage, we utilize the collected raw training data to train publicly released LMs to satisfy the requirement of automatically generating instruction-tuning data in certain privacy or copyright domains, which do not allow disclosure of data to ChatGPT. We train two LMs as a task generator $\mathcal{M}_g$ and a task discriminator $\mathcal{M}_d$, respectively. The former aims to design tasks based on the given text, while the latter evaluates the designed tasks in order to retain high-quality tasks. To accomplish these goals, we apply LLAMA2-7B (Touvron et al., 2023b), one of the most advanced LMs publicly available as the initialization of both generator and discriminator.

**TeG-Instruct Generator.** We first perform *supervised fine-tuning* (SFT) on $\mathcal{M}_g$ using the document-task pairs $\{(\mathcal{D}^+, \mathcal{T}^+)\}$ of $\Omega^+$. To adapt LLAMA2 to our defined SFT process, we add a meta-instruction $\mathcal{P}_g$ to each $(\mathcal{D}^+, \mathcal{T}^+)$, to describe the mapping from $\mathcal{D}^+$ to $\mathcal{T}^+$. In our experiments, the $\mathcal{P}_g$ is identical for all the training examples, $\mathcal{P}_g$ = "Convert the given text into a task. Input is a text and Response contains three fields: #instruction#, #input# and #output#.". In this way, the training loss is calculated by a log-likelihood, $\mathcal{L}(\mathcal{P}_g, \mathcal{D}^+, \mathcal{T}^+) = -\log P(\mathcal{T}^+ | \mathcal{P}_g, \mathcal{D}^+) = -\sum_{j=1}^{|\mathcal{T}^+|} \log P(t_j^+ | \mathcal{P}_g, \mathcal{D}^+, t_{<j}^+)$, where $t_j^+$ is the $j$-th token of $\mathcal{T}^+$ and $P(t_j^+ | \mathcal{P}_g, \mathcal{D}^+, t_{<j}^+)$ is the predicted probability at each step of the autoregressive decoding.

**TeG-Instruct Discriminator.** The sole goal of $\mathcal{M}_d$ is to perform a bi-classification that determines whether each task generated by $\mathcal{M}_g$ is valid. Each training example can be represented as $(\mathcal{D} \parallel \mathcal{T}, \mathcal{Y})$, where $\parallel$ represents the concatenation of two texts. In this setting, each $(\mathcal{D}^+, \mathcal{T}^+) \in \Omega^+$ is transformed to a positive example $(\mathcal{D}^+ \parallel \mathcal{T}^+, \mathcal{Y}^+)$, where $\mathcal{Y}^+$ = "valid", and each $(\mathcal{D}^-, \mathcal{T}^-) \in \Omega^-$ is converted to a negative example $(\mathcal{D}^- \parallel \mathcal{T}^-, \mathcal{Y}^-)$, where $\mathcal{Y}^-$ = "invalid". We add a meta-instruction $\mathcal{P}_d$ to help the model understand the target, $\mathcal{P}_d$ = "Given a piece of text and a task generated from that text, determine if the task is valid or invalid.". For each $(\mathcal{D} \parallel \mathcal{T}, \mathcal{Y})$, the training loss is calculated by a similar log-likelihood, $\mathcal{L}(\mathcal{P}_g, \mathcal{D} \parallel \mathcal{T}, \mathcal{Y})$. Note that $\mathcal{M}_d$ is necessary in our method because only a fraction of the invalid tasks can be recognized by the rule during the training phase, and the rest of the tasks need to ask ChatGPT. However, it is impractical to query ChatGPT for massive document-task pairs in the inference phase. Instead, this job is completed by $\mathcal{M}_d$.

Table 1: Statistics of $\mathcal{X}_d$, $\mathcal{X}_t$, $\Omega^+$ and TEG-INSTRUCT. Instruction, input and output lengths are given as the number of characters. Here $x \pm y$ denotes the average $x$ and standard deviation $y$.

| | # of Examples | Instruction Length | Input Length | Output Length |
|---|---|---|---|---|
| $\mathcal{X}_d$ | 189 | $146 \pm 72$ | $566 \pm 977$ | $548 \pm 464$ |
| $\mathcal{X}_t$ | 50 | $70 \pm 24$ | $74 \pm 107$ | $409 \pm 448$ |
| $\Omega^+$ | 7175 | $160 \pm 97$ | $485 \pm 854$ | $595 \pm 474$ |
| TEG-INSTRUCT | 274470 | $200 \pm 258$ | $568 \pm 971$ | $486 \pm 560$ |

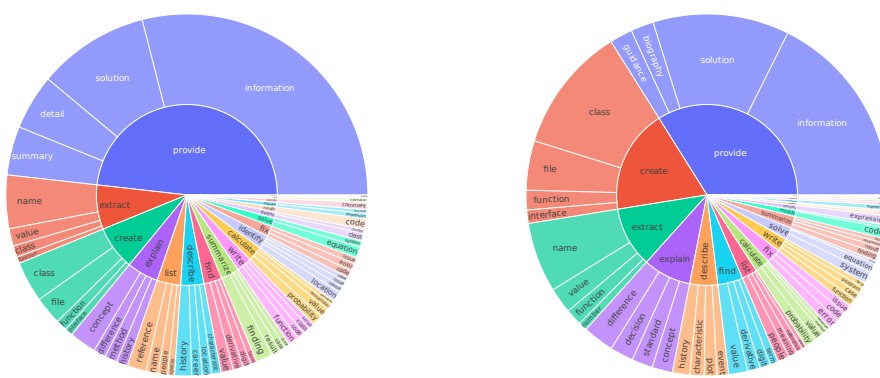

Figure 3: Instruction diversity of positive set $\Omega^+$ (left) and TEG-INSTRUCT data (right). The inner circle shows common root verbs with the corresponding common noun objects in the outer circle. The TEG-INSTRUCT data we generated using LLAMA2-7B is essentially on par in diversity with $\Omega^+$ generated by ChatGPT and even provides a more reasonable distribution.

## 4 TEG-INSTRUCT DATA STATISTICS

**Data Statistics.** Table 1 shows the statistics of the seeds, $\Omega^+$, and TEG-INSTRUCT dataset. Note that our method has no theoretical upper limit on the amount of data that can be generated as long as $\mathcal{M}$ produces continuously. Considering the experimental efficiency, we only generated 270k data. The instructions, inputs and outputs of $\mathcal{X}_d$ are significantly longer compared to the seed task $\mathcal{X}_t$ extracted from ALPACA-GPT4. TEG-INSTRUCT tends to have longer inputs and outputs compared to the seed data and $\Omega^+$. Notice that both $\Omega^+$ and TEG-INSTRUCT have large standard deviations regarding the input and output fields. This observation highlights that the tasks obtained from text-grounded designs may exhibit substantial variations in length. The top of Table 2 presents the statistical data for various corpora in TEG-INSTRUCT. FreeLaw exhibits the longest input length due to its generated tasks frequently involving the reading of extensive legal texts. Conversely, Wikipedia generally has the shortest input length, possibly because its tasks often lack input fields.

**Diversity of Instructions.** We perform a diversity analysis on both $\Omega^+$ and TEG-INSTRUCT using the method described by Wang et al. (2023). Figure 3 illustrates the distribution of the verb-noun structure of instructions in the meta-training set (left) and TEG-INSTRUCT (right), respectively. The TEG-INSTRUCT data generated using LLAMA2-7B exhibits a comparable level of diversity to the meta-training set created by ChatGPT. In fact, it offers a more reasonable distribution of data.

**Relevance to Documents.** Additionally, we have computed the relevance of both the inputs and outputs of the generated tasks to the original text. The relevance scores are displayed at the bottom of Table 2, with $\tilde{\sigma}$ representing the measure of literal relevance utilized in our post-processing, and BS denoting Bert-Score (Zhang et al., 2019) for evaluating the semantic relevance. In terms of literal relevance, it is noteworthy that the inputs and outputs of TEG-INSTRUCT's tasks predominantly originate from the provided text. The slight decrease in relevance scores for the output can be attributed to the model's tendency to condense and modify the original text as needed. The relatively lower BS scores, particularly for the input $\mathcal{I}$ related to Wikipedia and Arxiv, can be attributed to the fact that the input size for these tasks is significantly smaller compared to the given output $\mathcal{O}$.

Table 2: Statistics of different corpora in TEG-INSTRUCT. Instruction, input, and output lengths are given as the number of characters. BS denotes the Bert-Score(Zhang et al., 2019).

| | Wikipedia | FreeLaw | ArXiv | StackExchange | DM Math | Github |
|---|---|---|---|---|---|---|
| **# of Examples** | 33787 | 56863 | 29380 | 69706 | 14576 | 70158 |
| **Length of $\mathcal{P}_i$** | $71 \pm 27$ | $111 \pm 55$ | $99 \pm 47$ | $467 \pm 397$ | $49 \pm 19$ | $140 \pm 50$ |
| **Length of $\mathcal{I}_i$** | $2 \pm 24$ | $1207 \pm 1349$ | $74 \pm 422$ | $24 \pm 143$ | $44 \pm 32$ | $1178 \pm 900$ |
| **Length of $\mathcal{O}_i$** | $680 \pm 600$ | $627 \pm 581$ | $622 \pm 585$ | $463 \pm 439$ | $6 \pm 6$ | $344 \pm 581$ |
| $\tilde{\sigma}(\mathcal{D}_i, \mathcal{I}_i)$ | 1.000 | 0.997 | 0.999 | 0.998 | 0.951 | 0.934 |
| $\tilde{\sigma}(\mathcal{D}_i, \mathcal{O}_i)$ | 0.960 | 0.949 | 0.968 | 0.976 | 0.978 | 0.957 |
| $\text{BS}(\mathcal{D}_i, \mathcal{I}_i)$ | 0.080 | 0.814 | 0.454 | 0.915 | 0.874 | 0.910 |
| $\text{BS}(\mathcal{D}_i, \mathcal{O}_i)$ | 0.914 | 0.841 | 0.862 | 0.876 | 0.821 | 0.900 |

Table 3: Human evaluation of compared instruction-tuning datasets. For each dataset, we randomly sampled 50 examples. DYNOSAUR does not have the last four metrics because it directly uses the inputs and outputs from the previous datasets. LAMINI, WIZARDLM, LONGFORM, and HUMPBACK$^+$ do not have input field. Here $\downarrow$ means the smaller the value, the better.

| Dataset | $\text{CL}_\mathcal{P}$ (%) | $\text{HA}_\mathcal{I}$ (%) $\downarrow$ | $\text{HA}_\mathcal{O}$ (%) $\downarrow$ | $\text{FL}_\mathcal{I}$ (%) | $\text{FL}_\mathcal{O}$ (%) |
|---|---|---|---|---|---|
| SELF-INSTRUCT (Wang et al., 2023) | 92 | 28 | 32 | **96** | 92 |
| ALPACA (Taori et al., 2023) | 88 | 20 | 38 | **96** | 96 |
| ALPACA + GPT-4 (Taori et al., 2023) | 94 | 14 | 22 | 94 | 98 |
| UNNATURAL (Honovich et al., 2023) | 92 | 16 | 24 | **96** | 92 |
| DYNOSAUR (Yin et al., 2023) | **98** | - | - | - | - |
| LAMINI (Wu et al., 2023) | 94 | - | 20 | - | **98** |
| WIZARDLM (Xu et al., 2023) | 94 | - | 14 | - | **96** |
| LONGFORM (Köksal et al., 2023) | 76 | - | 10 | - | 84 |
| HUMPBACK$^\dagger$ (Li et al., 2023) | 48 | - | 12 | - | 64 |
| $\Omega^+$ | 92 | 12 | **10** | **96** | 94 |
| TEG-INSTRUCT | 94 | **10** | 12 | **96** | 96 |

## 5 EXPERIMENTS

### 5.1 EXPERIMENTAL SETUP

**Compared Datasets.** We compare TEG-INSTRUCT with several existing instruction-tuning datasets: SELF-INSTRUCT (Wang et al., 2023), ALPACA (Taori et al., 2023), ALPACA + GPT-4 (Taori et al., 2023), and UNNATURAL INSTRUCTIONS (Honovich et al., 2023) are automatically generated by LLMs including ChatGPT and text-davinci-002. LAMINI (Wu et al., 2023) is created by leveraging existing instructions, such as SELF-INSTRUCT and ALPACA, as its foundation. DYNOSAUR (Yin et al., 2023) repackages huggingface's existing NLP dataset and regenerates instructions for it using ChatGPT. LONGFORM (Köksal et al., 2023) and HUMPBACK (Li et al., 2023) are most similar to our work in that they generate tasks by performing instruction back translation on human-written texts. Here, since HUMPBACK hasn't been released yet, we got an unofficial version from HuggingFace [2], denoted by $\dagger$. The key distinction between our TEG-INSTRUCT and theirs lies in the way TEG-INSTRUCT wraps the text and carefully selects the essential components to compose a comprehensive task, incorporating the instruction, input, and output. This minimizes the noise present in the original text and provides a more streamlined and coherent task structure.

**Implementation Details.** All our experiments ran on 8 Tesla V100 GPUs with FP16. We trained $\mathcal{M}_g$ and $\mathcal{M}_d$ using LoRA (Hu et al., 2022). The hyper-parameters were set as follows: (1) The batch size of both $\mathcal{M}_g$ and $\mathcal{M}_d$ were set to 128. (2) The learning rate was set to $1 \times 10^{-4}$. (3) The epoch number was set to 7. (4) The LoRA target modules consisted of $[q_{proj}, k_{proj}, v_{proj}, o_{proj}, up_{proj}, down_{proj}, gate_{proj}, embed_{tokens}, lm_{head}]$. (5) The cutoff length was set to 2048. (6) The temperature and beam size of generation were 0.1 and 4, respectively.

---

[2] https://huggingface.co/datasets/Spico/Humback

Table 4: Human evaluation comparing TEG-INSTRUCT with various text-grounded instruction-tuning data collection methods, using identical input human-written text.

| Dataset | CHATGPT | LONGFORM | HUMPBACK[†] |
|---|---|---|---|
| *Our Task Designer Wins* (%) | 32.5 | 24.2 | 67.0 |
| *Tie* (%) | 35.5 | 63.1 | 28.2 |
| *Our Task Designer Loses* (%) | 32 | 12.7 | 4.8 |

Table 5: Rouge-L scores (%) for different document sources on the test set of TEG-INSTRUCT.

| | Wiki | FL | ArXiv | SE | DM | Github | Average |
|---|---|---|---|---|---|---|---|
| ALPACA (Taori et al., 2023) | 31.5 | 51.6 | 38.5 | 27.8 | 15.3 | 64.2 | 43.1 |
| DYNOSAUR (Yin et al., 2023) | 25.1 | 21.4 | 23.6 | 10.6 | 13.4 | 44.6 | 24.8 |
| WIZARDLM (Xu et al., 2023) | 26.5 | 44.1 | 32.7 | 30.8 | 10.3 | 43.5 | 34.5 |
| LONGFORM (Köksal et al., 2023) | 30.2 | 38.3 | 35.5 | 25.4 | 16.4 | 61.5 | 37.9 |
| HUMPBACK[†] (Li et al., 2023) | 26.5 | 43.1 | 35.7 | 30.3 | 11.4 | 46.9 | 36.4 |
| $\Omega^{+}$ | 29.8 | 36.1 | 34.5 | 27.8 | 12.7 | 47.5 | 31.3 |
| TEG-INSTRUCT | **38.0** | **61.9** | **42.3** | **36.4** | **45.3** | **79.7** | **53.7** |

## 5.2 HUMAN EVALUATION

We first randomly select 50 examples from each dataset and manually evaluate their quality. Since the generated tasks may involve knowledge from several different domains, we require that the annotator needs to retrieve the corresponding evidence using the search engines and compare them one by one. The entire process of manual evaluation took approximately 40 man-hours.

**Evaluation Metrics.** a) *instruction clarity* ($CL_{\mathcal{P}}$) indicates the percentage of instructions that are correct and make sense. b) *input hallucination* ($HA_{\mathcal{I}}$) and *output hallucination* ($HA_{\mathcal{O}}$) measure how much input and output contain factual or logical errors, respectively. c) *input fluency* ($FL_{\mathcal{I}}$) and *output fluency* ($FL_{\mathcal{O}}$) gauge the extent to which the input and output exhibit fluency and adhere to the dialog scenario, excluding any extraneous information.

**Main Results.** The results are shown in Figure 3. SELF-INSTRUCT, ALPACA, ALPACA + GPT-4, LAMINI, and UNNATURAL INSTRUCTIONS leverage the robust language generation capabilities of LLMs, resulting in their instructions demonstrating high $CL_{\mathcal{P}}$, $FL_{\mathcal{I}}$, and $FL_{\mathcal{O}}$ scores. However, since they lack the support of actual text, these models are susceptible to the problem of hallucination, thereby leading to higher $HA_{\mathcal{I}}$ and $HA_{\mathcal{O}}$ scores. DYNOSAUR likely attained the highest score due to being built upon extensively standardized datasets with a well-defined structure and specifications. By generating tasks from human-written text, both LONGFORM and HUMPBACK effectively mitigate the issue of hallucination in the output. Regrettably, the presence of noise in real text diminishes its fluency ($FL_{\mathcal{I}}$ and $FL_{\mathcal{O}}$) compared to fully self-generated datasets. Furthermore, our findings indicate that employing back-translation may lead the model to engage in dialog rather than generating clear instructions. In contrast, our dataset, TEG-INSTRUCT, encompasses the entire task design process based on the provided text. This approach not only mitigates the illusion of the task but also guarantees the coherence and smoothness of the instructions, as well as the input-output alignment. Additionally, it is crucial to highlight that TEG-INSTRUCT achieves its results using a 7B-sized model, demonstrating its efficiency and high scalability.

**Task Quality for the Same Document.** To demonstrate the superiority of our task designer for collecting text-grounded instruction-tuning data, we utilize it to generate tasks using documents from other text-grounded datasets and subsequently compare the results. Specifically, we each randomly sample 100 documents used in LONGFORM and HUMPBACK to feed our $\mathcal{M}$. The results are shown in Table 4. For the same batch of documents, our method generates more tasks of better quality. In addition, we randomly sample approximately 100 documents utilized in TEG-INSTRUCT and have ChatGPT convert them to tasks. Table 4 reveals that the tasks generated by our task designer are of similar quality to those generated by ChatGPT.

Table 6: Rouge-L (R), Meteor (M), and Bert-Score (B) of different methods on the test sets of out-of-domain benchmarks. All methods follow zero-shot settings.

| | RecipeNLG | | | ELI5 | | | Super-NI | | |
|---|---|---|---|---|---|---|---|---|---|
| | R(%) | M(%) | B(%) | R(%) | M(%) | B(%) | R(%) | M(%) | B(%) |
| SELF-INSTRUCT (Wang et al., 2023) | 24.8 | 25.9 | 81.7 | 9.8 | 8.2 | 81.3 | 23.1 | 20.8 | 83.0 |
| ALPACA (Taori et al., 2023) | 26.6 | 27.0 | 84.6 | 10.1 | 8.8 | 82.8 | 20.3 | 21.2 | 82.9 |
| ALPACA + GPT-4 (Taori et al., 2023) | 26.3 | 29.7 | 83.5 | 11.1 | 13.3 | 83.0 | 26.1 | 23.2 | 86.0 |
| UNNATURAL (Honovich et al., 2023) | 30.3 | 27.4 | 86.0 | 10.0 | 6.2 | 82.4 | **31.0** | **25.4** | **88.0** |
| DYNOSAUR (Yin et al., 2023) | 24.5 | 19.3 | 83.7 | 3.1 | 1.5 | 80.0 | 30.0 | 22.1 | 87.7 |
| LAMINI (Wu et al., 2023) | 30.2 | 29.1 | 83.8 | 14.5 | 9.8 | 83.9 | 30.7 | 25.1 | 87.2 |
| WIZARDLM (Xu et al., 2023) | 30.3 | **30.3** | 84.1 | **20.2** | **19.3** | 83.5 | 23.9 | 22.5 | 84.5 |
| LONGFORM (Köksal et al., 2023) | 28.1 | 26.4 | 86.0 | 7.5 | 5.4 | 78.7 | 14.4 | 16.6 | 79.9 |
| HUMPBACK[†] (Li et al., 2023) | 23.9 | 22.9 | 82.2 | 9.3 | 6.1 | 83.0 | 12.5 | 15.8 | 82.7 |
| $\Omega^+$ | 24.5 | 21.2 | 83.2 | 12.7 | 8.9 | 81.0 | 17.4 | 18.3 | 82.9 |
| TEG-INSTRUCT | **31.0** | 24.6 | **86.4** | 16.6 | 11.5 | **84.5** | 19.8 | 20.6 | 85.5 |

## 5.3 AUTOMATIC EVALUATION

To further evaluate the quality of these instruction-tuning datasets, we independently train an identical LLM on each dataset and observe its performance in following diverse instructions after training.

**Baseline LLM.** Consistent with $\mathcal{M}_g$ and $\mathcal{M}_d$, we choose LLAMA2-7B (Touvron et al., 2023b) + LORA (Hu et al., 2022) as the baseline LLM. For ease of presentation, we refer to the LLAMA2-7B+LORA trained on dataset $x$ as $x$-model.

**Benchmarks.** We first evaluate all the models on the test set of TEG-INSTRUCT, which consists of 10000 tasks, to evaluate the in-domain generation capability of our TEG-INSTRUCT-model. To assess the out-of-domain generalization capability of the trained LLMs, we conduct evaluations on three diverse benchmarks: Recipe Generation from Ingredients (RecipeNLG) (Bien et al., 2020), Long-Form Question Answering (ELI5) (Fan et al., 2019), and Super-NaturalInstructions (Super-NI) (Wang et al., 2022). To ensure the experiment's efficiency of RecipeNLG, we randomly selected 20,000 examples from it as a test set. It is worth emphasizing that none of the instruction tuning datasets used include data for these benchmarks (zero-shot).

**Evaluation Metrics.** Following (Köksal et al., 2023; Yin et al., 2023), we present the Rouge-L (Lin, 2004) and Meteor (Banerjee & Lavie, 2005) scores for the model outputs with the human annotations in the two datasets. We evaluated Bert-Score (Zhang et al., 2019) to serve as a semantic similarity between the prediction and the gold label.

### 5.3.1 IN-DOMAIN RESULTS.

Table 5 shows the Rouge-L scores of different models on the test set of TEG-INSTRUCT. Compared to the baseline model, the performance of TEG-INSTRUCT-model shows a significant improvement, especially in terms of the ability to code relevant tasks (Github). The DYNOSAUR model exhibits the poorest performance, possibly because its data is overly standardized, whereas our TEG-INSTRUCT model closely resembles human-written text. Additionally, most of the other baselines are not suitable for our dataset due to the absence of necessary input fields, except for ALPACA. Interestingly, despite the close quality of $\Omega^+$ and TEG-INSTRUCT, the $\Omega^+$-model does not achieve a significant advantage, suggesting that the amount of instruction-tuning data also affects LLMs' performance.

### 5.3.2 OUT-OF-DOMAIN RESULTS.

Table 5 shows the out-of-domain generalization capability of different models on the test set of the three benchmarks. Our TEG-INSTRUCT-model achieves the best Bert-Score on Eli5 and the best Rouge-L and Bert-Score on RecipeNLG. The literal performance (Rouge-L and Meteor) of the model is lower compared to its semantic performance (Bert-Score). We speculate that this discrepancy arises from the varied outputs of the models, which may not align precisely with the gold annotations. In contrast to the first two datasets, our TEG-INSTRUCT-model demonstrates lower performance on Super-NI. We observed that the initial datasets primarily consisted of lengthy texts

as standard outputs, whereas Super-NI predominantly consisted of categorization tasks with shorter output requirements. Therefore, the variation in task distribution could be the primary factor contributing to the inadequacy of our methodology.

# 6    RELATED WORK

**Instruction Tuning** Humans possess the ability to effortlessly comprehend and execute tasks based on verbal instructions (Touvron et al., 2023a; OpenAI, 2023; Touvron et al., 2023b). Likewise, advancements in deep learning have enabled Language Models (LLMs) (Brown et al., 2020; OpenAI, 2023; Chowdhery et al., 2022; Touvron et al., 2023a) to acquire the capability to understand and follow instructions. Instruction tuning serves as a promising method, involving the fine-tuning of LLMs using training data and instructions from a collection of upstream tasks(Sanh et al., 2022; Mishra et al., 2022; Wei et al., 2022; Chung et al., 2022; Longpre et al., 2023; Peng et al., 2023). Subsequently, these models can then be employed to perform inference on unfamiliar tasks using both instructions and instance inputs. In this paper, we not only train instruction tuning models but also propose a novel method for formulating training exmaples.

**Instruction-Tuning Data Collection** The collection of high-quality instruction-tuning data (Xu et al., 2023; Yin et al., 2023; Honovich et al., 2023) is a pressing issue in enhancing the capability of instruction-following. Previous approaches can be broadly categorized into three main groups. Firstly, methods like SUPER-NI (Wang et al., 2022) and DOLLY (Conover et al., 2023) rely on hiring professionals to create instructions for diverse NLP tasks. However, these methods suffer from limited scalability due to the labor-intensive nature of the process. Secondly, approaches such as SELF-INSTRUCT (Wang et al., 2023) and ALPACA (Taori et al., 2023) advocate for the use of LLMs to automatically generate instruction-tuning data, thus eliminating the need for manual labor. However, the data quality is compromised due to the inherent limitations and biases of LLMs. Lastly, Dynosaur (Yin et al., 2023) employs LLMs to convert existing NLP datasets from platforms like HuggingFace into instruction-tuning data at a reduced cost. Unfortunately, this approach is not applicable in scenarios where no dataset is available.

All of the mentioned approaches utilize model-generated responses as training data. However, a method closely related to ours is the simultaneous work by (Köksal et al., 2023; Li et al., 2023). Their approach involves using human-written text as a natural response and leveraging an LLM to generate the corresponding instruction based on the response. The primary differentiation between our TEG-INSTRUCT method and theirs is the approach we take to encapsulate the text and meticulously select the crucial elements for constructing a comprehensive task. In our released TEG-INSTRUCT, we integrate the instruction, input, and output, carefully curating these components to minimize the noise inherent in the original text. This process results in a more streamlined and coherent structure for the task at hand.

# 7    CONCLUSION

This paper presented a scalable method for automatically gathering high-quality instruction-tuning data, aiming to enhance the capabilities of LLMs. Our method involves training open-source LLMs to design tasks based on human-written texts, which are used to effectively address the issue of hallucinations in generated responses. Unlike other text-grounded methods, our method requires the LLM to simultaneously generate the instruction, input, and output, thereby minimizing noise from the original text and maintaining the fluidity of the generated tasks. Building upon this approach, we released a dataset comprising 270,000 instruction-tuning examples. We have conducted automated and manual evaluation experiments to assess the quality of our dataset. The results demonstrate the effectiveness of our approach in producing high-quality data. The LLMs trained from our dataset exhibit competitive in-domain and out-of-domain generation capabilities.

Our proposed method offers an efficient solution for collecting reliable instructional adaptation data. By leveraging language models and human-written texts, we enhance LLM capabilities and provide a benchmark for future research in instruction-tuning and data collection methodologies. In future work, we will explore generating more complex tasks that involve multiple documents. This can be achieved by federating or aggregating information from multiple sources to create richer and more comprehensive instructional tasks.

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
