# OpenReview forum: "TeG-Instruct: Towards Premium Instruction-Tuning Data via Text-Grounded Task Design"
_ICLR.cc/2024/Conference — ICLR 2024 Conference Withdrawn Submission_

### Official Review · Reviewer_kxN5 · 2023-10-28

**Soundness:** 3 good
**Presentation:** 3 good
**Contribution:** 2 fair
**Rating:** 5
**Confidence:** 3

**Summary:**

This paper presents *Teg-Instruct*, a scalable method for collecting instruction-tuning data that incorporates the instruction, input, and output based on the given human-written text. Firstly, a meta-training set was built by leveraging well-aligned teach models, specifically, ChatGPT. The meta-training set is further used for training two LMs, one for generating instruction, input, and output given documents, and the other for evaluating the generated content as a bi-classification problem. Experiments are conducted both in-domain and out-domain using the collected 270k samples with Llama2-7B and indicate the effectiveness of the proposed dataset. Also, a comprehensive analysis of automatic and human analysis is done to showcase the quality of the constructed *Teg-Instrct*.

**Strengths:**

- The proposed method is scalable and the released dataset is large-scale.
- The paper is generally well-written and organized. Almost everything is clear to me.
- Thorough automatic and human evaluations over a range of instruction-tuning datasets on various downstream tasks. The analysis for the curated *Teg-Instruct* is also comprehensive.

**Weaknesses:**

- I'm not so convinced by the experiment results. It is not surprising that *Teg-Instruct* achieves the best performance for in-domain tests. For out-of-domain experiment results, the performance is not that satisfactory. It. is understandable for different output distributions. Instead, what is the performance on some general benchmarks aside from the in-domain and out-domain evaluations, such as LongForm and Alpaca Eval?

- Not a strict weakness, just wondering why it is called "text-grounded task design"? I'm a little confused by the term "task design". In my view, "task" usually refers to well-defined task formats, such as QA, IE, etc.

**Questions:**

1. Could you further explain the score $\sigma$ used in post-processing?

2. What are the thresholds used in post-processing, such as $\theta$? And how are they decided? Is $\theta$ the same for the two similarity calculations?

3. What is the detailed setting for human evaluation? For example, how many people are involved in the annotation process? What are their backgrounds? Since the number of examples for evaluation is relatively small, i.e., 50, how reliable is the human evaluation?

---

### Official Review · Reviewer_5ypF · 2023-10-29

**Soundness:** 3 good
**Presentation:** 2 fair
**Contribution:** 2 fair
**Rating:** 5
**Confidence:** 4

**Summary:**

This paper proposes a method to generate instruction-tuning datasets grounded on corpus, aiming to address the problems like hallucination. The dataset generated contains 270k examples. The generating process contains 2 stages: (1) collect seed examples with the help of LLMs; (2) train language models to generate more examples and filter out low-quality ones. In the test set proposed in this paper (TEG-Instruct test set) and on several other benchmarks, the method presented in this paper performs well.

**Strengths:**

- This paper proposes a text-grounded instruction-tuning dataset with up to 270k examples, and the method is scalable.
 - The method trains a discriminator to filter out low-quality data, which can alleviate the problem of hallucination.
 - The motivation of this paper is intuitive.

**Weaknesses:**

- The contribution of this paper is a little bit limited, and the differences compared to [1] and [2] are not very distinct.
 - There are some concerns regarding the evaluation of the proposed method and dataset. In terms of automated evaluation, using the test set generated in this paper, TEG-Instruct, as the testing dataset might be somewhat unfair. Additionally, including evaluation results on more benchmarks can enhance the robustness of the article.

[1] "Self-Alignment with Instruction Backtranslation". Li et al.
[2] "Longform: Optimizing instruction tuning for long text generation with corpus extraction." Köksal et al.

**Questions:**

- What is the difference between this work and other text-grounded instruction-tuning dataset generation methods like [1] and [2]?
 - How does the paper demonstrate the alleviating of the problems of hallucination in the evaluation?
 - Are there any ablation experiments to demonstrate that the dataset format with (Instruction, Input, Output) proposed in the paper is better than the format with (Instruction, Output), particularly in instruction tuning?

[1] "Self-Alignment with Instruction Backtranslation". Li et al.
[2] "Longform: Optimizing instruction tuning for long text generation with corpus extraction." Köksal et al.

---

### Official Review · Reviewer_q6hF · 2023-11-01

**Soundness:** 3 good
**Presentation:** 2 fair
**Contribution:** 2 fair
**Rating:** 3
**Confidence:** 4

**Summary:**

The paper introduces Teg-Instruct, a paradigm for collecting instruction-tuning data. The procedure consists of two steps: First, well-aligned LLMs are leveraged to generate seed data from both document-view and task view. Then, the seed data is used to train two LMs as a task generator and a task discriminator. The paper provides both human evaluation and automatic evaluation on the generated Teg-Instruct dataset, demonstrating that Teg-Instruct is of high quality and improves both in-domain and out-of-domain instruction following abilities for LLMs.

**Strengths:**

+ The paper provides extensive and thorough experimentation on the data quality. Particularly, The human evaluation on the intrinsic quality of the instruction tuning datasets is valuable.

+ The instruction tuning datasets will be released and will be valuable to the community.

**Weaknesses:**

- The improvement from the dataset seems incremental compared with previous datasets and methods especially in terms of downstream evaluation with out-of-domain data.

- The data creation process is complicated with many rules not clearly specified.

**Questions:**

It is not entirely clear how the Teg-Instruct seed data was constructed. It was mentioned that both document-view and task-view data in the first stage were manually checked. What is the manual review process? Also, what is the similarity threshold in the post processing for removing hallucination examples?

---

### Official Review · Reviewer_byqM · 2023-11-01

**Soundness:** 2 fair
**Presentation:** 3 good
**Contribution:** 1 poor
**Rating:** 3
**Confidence:** 4

**Summary:**

This article proposes TeG-Instruct, which uses a specially trained generator to generate training samples for instruction tuning and trains a discriminator to filter the samples generated by the generator. This method can generate more training samples for instruction tuning at a lower cost.

**Strengths:**

The introduced method can generate more Instruction tuning samples at a lower cost.

**Weaknesses:**

1. The paper states in Figure 1 that human-written text contains noise. However, this is a rare case. If the authors think this is a common problem, then all the human-annotated tasks are pointless. The author uses examples generated by ChatGPT as training samples to train a Llama-2-7B model to generate new training samples. I think the training samples generated in this case will have more noise.
2. The paper states in Figure 1 that the lack of input field is detrimental to LLM’s generalization capability in diversity tasks. There is no verification of it, no explanation in other papers, and no ablation experiments.
3. This method has almost no improvement in out-of-domain datasets compared to other baselines, and the in-domain experiment is almost meaningless because other baselines are all in the setting of out-of-domain when dealing with the dataset proposed in this paper. This is an unfair comparison. In addition, the article lacks the number of training data for other baselines.

**Questions:**

N/A

---

### Meta-Review · Area_Chair_429y · 2023-12-04

**Metareview:**

The paper introduces TeG-Instruct, a new method for generating instruction-tuning data at a lower cost. The main novelty lies in the combination of a generator trained on seed data from well-aligned LLMs and a discriminator to filter out low-quality samples. Experiments are conducted on both in-domain and out-of-domain data. On the positive side, the proposed approach appears to be scalable, and the released dataset could be valuable to the community. However, the reviewers had several serious concerns about accepting this paper. First, the novelty of this work seems to be relatively limited compared to prior works on instruction tuning motivated in similar ways (see reviews 5ypF and q6hF for details, and review 5ypF for references). Second, positive results on in-domain data are seldom surprising (as conducted in TeG-Instruct's test set), while out-of-domain results are relatively mixed. Finally, using TeG-Instruct's own test set to compare the approach of the paper against prior work is somewhat unfair. The authors did not respond to the reviewers to address these concerns. In light of these limitations, I recommend rejecting this paper.

**Justification For Why Not Higher Score:**

The reviewers express concerns regarding the limited novelty of the paper and the utilization of test data generated by their approach to evaluate their work in comparison to prior research.

**Justification For Why Not Lower Score:**

N/A

---

### Decision · Program_Chairs · 2024-01-16

Reject